# Serovars, Virulence and Antimicrobial Resistance Genes of Non-Typhoidal *Salmonella* Strains from Dairy Systems in Mexico

**DOI:** 10.3390/antibiotics12121662

**Published:** 2023-11-25

**Authors:** Stephany Barrera, Sonia Vázquez-Flores, David Needle, Nadia Rodríguez-Medina, Dianella Iglesias, Joseph L. Sevigny, Lawrence M. Gordon, Stephen Simpson, W. Kelley Thomas, Hectorina Rodulfo, Marcos De Donato

**Affiliations:** 1Tecnologico de Monterrey, School of Engineering and Sciences, Querétaro 76130, CP, Mexico; stephbarrera@tec.mx (S.B.); diglesias@tec.mx (D.I.); herodulfo@tec.mx (H.R.); 2Veterinary Diagnostic Lab, University of New Hampshire, Durham, NH 03824, USA; david.needle@unh.edu; 3Instituto Nacional de Salud Pública (INSP), Centro de Investigación Sobre Enfermedades Infecciosas (CISEI), Cuernavaca 62100, MR, Mexico; nadia_yeli@hotmail.com; 4Department Molecular, Cellular and Biomedical Sciences, University of New Hampshire, Durham, NH 03824, USA; jlsevigny1@wildcats.unh.edu (J.L.S.); lawrence.gordon@unh.edu (L.M.G.); stephen.simpson@unh.edu (S.S.); kelley.thomas@unh.edu (W.K.T.); 5The Center for Aquaculture Technologies, San Diego, CA 92121, USA

**Keywords:** *Salmonella*, dairy farms, periparturient cows, calves, maternity beds, virulence genes, antimicrobial resistance genes

## Abstract

*Salmonella* isolated from dairy farms has a significant effect on animal health and productivity. Different serogroups of *Salmonella* affect both human and bovine cattle causing illness in both reservoirs. Dairy cows and calves can be silent *Salmonella* shedders, increasing the possibility of dispensing *Salmonella* within the farm. The aim of this study was to determine the genomic characteristics of *Salmonella* isolates from dairy farms and to detect the presence of virulence and antimicrobial resistance genes. A total of 377 samples were collected in a cross-sectional study from calves, periparturient cow feces, and maternity beds in 55 dairy farms from the states of Aguascalientes, Baja California, Chihuahua, Coahuila, Durango, Mexico, Guanajuato, Hidalgo, Jalisco, Queretaro, San Luis Potosi, Tlaxcala, and Zacatecas. Twenty *Salmonella* isolates were selected as representative strains for whole genome sequencing. The serological classification of the strains was able to assign groups to only 12 isolates, but with only 5 of those being consistent with the genomic serotyping. The most prevalent serovar was *Salmonella* Montevideo followed by *Salmonella* Meleagridis. All isolates presented the chromosomal *aac(6′)-Iaa* gene that confers resistance to aminoglycosides. The antibiotic resistance genes *qnr*B19, *qnr*A1, *sul*2, *aph*(6)-Id, *aph*(3)-ld, *dfr*A1, *tet*A, *tet*C, *flor*2, *sul*1_15, *mph*(A), *aad*A2, *bla*CARB, and *qac*E were identified. Ten pathogenicity islands were identified, and the most prevalent plasmid was Col(pHAD28). The main source of *Salmonella enterica* is the maternity areas, where periparturient shedders are contaminants and perpetuate the pathogen within the dairy in manure, sand, and concrete surfaces. This study demonstrated the necessity of implementing One Health control actions to diminish the prevalence of antimicrobial resistant and virulent pathogens including *Salmonella.*

## 1. Introduction

*Salmonella enterica* is one of the most important infectious pathogens in humans and animals worldwide. These bacteria are ubiquitous in animals, humans, and the environment, facilitating their transmission. Currently, *S. enterica* comprises at least 2600 different serovars [1,2] and *S. enterica* subsp. *enterica* represents the most prevalent sub-species related to human and animal infections, accounting of at least 1500 serovars [1,3]. In food-producing animals, such as dairy cattle, the increasing prevalence of *Salmonella enterica* has become a major public and animal health issue [4]. Infected cattle can shed *Salmonella* into the environment for prolonged periods after subclinical infections [5], which can survive outside the host, perpetuating in-herd transmission [6]. 

The Center for Disease Control and Prevention (CDC) considers *Salmonella enterica* a serious threat, due to its virulence and its capacity to carry antibiotic resistance genes (ARG) [7]. CDC and World Health Organization (WHO) suggests immediate action for healthcare providers and veterinarians working with animal owners and producers to document antibiotic use data and monitoring antimicrobial resistance in foodborne bacteria for identifying emerging resistance. Henceforth, these practices could lead to the development and assessment of mitigation strategies of antimicrobial resistance [8]. The importance of the analysis of ARG in *S. enterica* is based on the characteristic of its transmission to humans through the food chain such as poultry and dairy products [9,10]. It is well documented that *Salmonella* enterica serovar Enteritidis is linked to infections in poultry and humans around the world [11,12,13]. Additionally, this serovar has the capacity to acquire antibiotic resistance genes, detecting multidrug-resistant isolates from poultry farms worldwide (91.1%) [14]. To tackle this problem, the One Health approach considers surveillance programs to detect ARG isolated from animals and the environment [15] to prepare for current and future zoonotic outbreaks [16]. 

In Mexico, there is little information about the prevalence of *Salmonella enterica* in apparently healthy cattle and farm environments, the diversity of serovars associated with livestock, and the presence of antimicrobial resistance and virulence genes [17,18]. Therefore, the objectives of this study were to determine the prevalence of *Salmonella enterica* in dairy systems, examine the serovars and sequence types, and evaluate the presence of virulence and antimicrobial resistance genes circulating in 13 states of Mexico. 

## 2. Results

### 2.1. Prevalence of Salmonella in the Different Samples

A total of 55 dairy herds were enrolled in this study from 13 states in Mexico, with a median herd size of 1570 cows (range: 200–10,000). From the collected samples (*n* = 377), 57 were positive for *Salmonella* through biochemical testing. The isolates were all motile, positive for H_2_S, and negative for indole. Of the 55 sampled dairies, 35 had at least one positive result, with a 63.6% prevalence. Stratified analysis indicated that 10 calves (10/127) were positive from 7 to 49 days of age in eight dairies, 16 periparturient cows (16/116) in 14 dairies, and 31 environmental isolates (31/134) from maternity areas, accounting for 25 positive dairies (*p* < 0.05). The highest number of isolates were found in the region called La Laguna (Coahuila and Durango), with 42% of the positive samples.

The PCR amplification of the *ipaB* gene produced the product of 315 bp in all the 57 isolated *Salmonella* strains.

### 2.2. Antimicrobial Susceptibility Testing 

The *Salmonella enterica* isolates were analyzed for antibiotics susceptibilities to five antibiotics using the disc diffusion method. Ampicillin and gentamicin resistance were identified in 10% of the isolates. Isolates showed intermediate resistance to ciprofloxacin, levofloxacin, amikacin, and gentamicin, with a frequency of 20, 70, 15, and 5%, respectively. None of the isolates showed a multidrug resistance pattern to more than two classes of antibiotics. 

Among the isolates that showed resistance, two were isolated from periparturient cows and one from calves and maternity beds each, all from different farms and geographical areas. 

### 2.3. Serological Identification of Salmonella enterica O Serogroups 

Twenty out of the fifty-seven isolates showed high or intermediate resistance against at least one antibiotic and were selected for further analysis. Of the 20 isolates, 12 could be classified in a serogroup level using the slide agglutination technique according to the Kaufmann–White–Le Minor method [1]. Three isolates were assigned to serogroup B, eight to C, and one to D. Serogroup A was not detected in this study. Only five isolates were consistent with the results obtained via genome sequencing, thus 59% of the serotyped isolates were misclassified. Differences in serogroup classification are shown in Table 1.

### 2.4. Genomic Characteristics

The sequencing produced 554.7 million clusters that passed the quality filtering, generating a total of 278. 5 Gbp of sequence data, of which 90.1% had a quality score of Q30 or higher (average of 35.2). The draft genome size of the *S. enterica* isolates ranged from 4.5 to 4.9 Mb, with a GC content from 50.0 to 52.3% (Appendix A). The number of contigs ranged from 40 to 200 and the N50 from 72.4 to 728.1 Kbp. The sequences were uploaded to GenBank under the BioProject PRJNA1018581 into the Short Reads Archive (SRA).

#### 2.4.1. In Silico Serovar Identification

Ten different serovars were identified in this study. *Salmonella* Montevideo was the most frequent serovar (*n* = 4, 20%) followed by Meleagridis (*n* = 3, 15%), Anatum (*n* = 3, 15%), Give (*n* = 2, 10%), Cubana (*n* = 2, 10%), London (*n* = 2, 10%), Newport (*n* = 1, 5%), Reading (*n* = 1, 5%), Agona (*n* = 1, 5%), and Havana (*n* = 1, 5%).

Among the ten serovars, four were identified in more than one state and six in one state only. *Salmonella* Give were observed in three states. The most heterogeneous distributions of serovars were observed in maternity beds where 9 out of 10 serovars were identified, *Salmonella* Newport, the only serovar not identified in the environment, was isolated from calf feces.

#### 2.4.2. In Silico Detection of Antimicrobial Resistance Genes

All 20 isolates carried at least one antimicrobial resistance gene. All of them encoded the chromosomal aminoglycoside acetyl-transferase gene *aac(6′)-Iaa*, associated with aminoglycoside resistance (Table 2 and Figure 1).

*Salmonella* serovar Newport presented the most extensive resistance profile, identifying the genes *tet(A)_6*, implicated in the resistance to tetracycline; *dfrA1_8*, associated with trimethoprim resistance; *floR_2*, with florfenicol resistance; *sul1*, implicated in sulfonamide resistance; *qnrA1_1*, associated with quinolone resistance; *bla*_CARB_, associated with class 1 integron-borne cassette that confers resistance to carbapenems; and *qacE*, related to quaternary ammonium compound resistance. Other antimicrobial resistance genes identified were *fosA*, a glutathione S-transferase that inactivates fosfomycin, presented in all isolates from serovars Meleagridis, Reading, and Agona. In 19 of the 20 isolates, *par*C T57S mutation was detected.

Four of the most prevalent serovars identified in this study were selected to determine the presence of determinants associated with antibiotic resistance, where 98 different genes were found in serovars Give, Montevideo, Meleagridis, and Anatum. Twenty-seven out of ninety-eight genes were identified only in the Montevideo serovar, and one was found in the Meleagridis serovar (Figure 1).

#### 2.4.3. Plasmid Profiling

Plasmids harbor several antimicrobial resistance genes that can contribute to ARG dissemination. Interestingly, only four out of twenty isolates were predicted to carry known plasmids (Table 3), and two of them were isolated from calf feces and the other two from maternity beds.

#### 2.4.4. In Silico Identification of Virulence Genes

The fimbrial adherence operons *fim*, *stb*, *std*, *sth*, *sti*, and *inv* (the type III secretion system 1 (T3SS-1) genes) and *sip*, *sop*A, *sop*D, *sop*E2, and *spa*OPQRS (the type III secretion system 2 (T3SS-2) genes) as well as *sif*AB and *ssc*AB were common in all the isolates. The bacteriophage-encoded ssph2 gene was identified in 13 isolates and the phage-related gene *sse*I was present only in serovar Newport. Of the four genes in the *pef*ABCD operon, only *pef*B was identified in 10 out of the 20 isolates, corresponding to *Salmonella* serovars Give, Anatum, Havana, Montevideo, and Cubana. The genes encoding typhoid toxin, *cdt*B, and *plt*A were identified only in the Salmonella serovars Give and Montevideo, and the *plt*B gene was present in one Montevideo isolate. The prophage-encoded gene *gog*A was present only in the Salmonella serovar Newport. Strains belonging to the same serovar exhibited a distinct virulence profile (Figure 2).

The same four serovars selected for the analysis of antibiotic resistance determinants were used to ascertain the presence of virulence genes linked to a specific serovar. A total of 364 virulence genes were detected in the four serovars, 28 were identified only in Montevideo, 8 in Meleagridis and Anatum, and none were specific to Give (Figure 3).

#### 2.4.5. In Silico Identification of *Salmonella* Pathogenicity Island

The most conserved *Salmonella* pathogenicity islands (SPI) were SPI-1, 2, 3, 4, 5, and 9, detected in all the isolates. SPI-8 was identified in 7 out of 20 isolates (35%), associated with the three *S.* Meleagridis and two *S.* Cubana, Havana, and Agona. Half of the isolates analyzed presented the SPI-13 and SPI-14, both detected in the same isolates, corresponding to *S*. Give, London, Newport, Anatum, Reading, and Montevideo. The CS54 pathogenicity island was detected in four isolates, corresponding to one *S*. Give, Reading, London, and Newport. C63PI was present in 12 isolates, associated with *S*. Give, London, Newport, Anatum, Agona, Meleagridis, Havana, and Cubana.

### 2.5. Phylogenetic Diversity

The core genome of SNPs of non-typhoidal *Salmonella* isolates were used to construct a phylogenetic tree to address its diversity. Our collection of isolates was distributed in three major clades, as seen in Figure 3. We observed that each clade was composed of isolates with different STs and serovars. Clade I was structured in isolates from Give and Montevideo serovars and three different STs. Clade II was composed of isolates from Meleagridis, Havana, and Agona serovars, while five STs were found within clade II. Isolates from clade III were distributed in six STs and four serovars: Anatum, Newport, Reading, and London.

MLST analysis identified 14 different sequence types among the 20 isolates. Four serovars exhibited more than one ST, such as Montevideo (ST 138 and 2269), Anatum (ST 64 and 1549), Meleagridis (ST 13 and 463), and London (ST 155 and 654). Strains from clade III belonging to ST 132, 1628, 654, and 155 show a distinct genotypic antimicrobial profile.

## 3. Discussion

*Salmonella enterica* has been described as a major cause of foodborne illness in humans and possess an important risk for animal health [3,19]. Important gastrointestinal bacteria, including *Salmonella,* enterohemorrhagic *Escherichia coli*, and *Listeria* spp., have been associated with outbreaks involved in contaminated cattle products [20]. The ability of *Salmonella* to establish asymptomatic infections in cattle could result in the intermittent shedding of this bacteria into the environment, challenging its control [21].

In the present study, a prevalence of 15% of *Salmonella enterica* was found in the dairy cattle and farm environment. Currently, information about the prevalence of *Salmonella* in dairy farms in Mexico is scarce. Some reports in cattle slaughtering and deboning operations found a frequency of 34% in fecal samples [22]. According to a meta-analysis of published studies of the prevalence and serotype diversity of *Salmonella* in apparently healthy cattle in USA, the overall pooled prevalence was 10% (95% CI: 7–13%) [23]. In another study conducted in 2007, the National Animal Health Monitoring System reported an overall prevalence of 13.7% of the total fecal samples analyzed, and in 30 of the 97 dairy herds in USA involved in the study had at least one *Salmonella*-positive fecal sample, representing 31% of positive herds; the results were based in a single sampling visit to each herd [24].

The highest frequency of *Salmonella enterica* was found in maternity beds. In most herds enrolled in this study, the maternity areas are shared for more than one periparturient cow, which possibly increases the prevalence of *Salmonella*. Environmental contamination can be the result of intermittent *Salmonella* shedding from asymptomatic periparturient cows, where bacteria can survive outside the host in suitable conditions for prolonged periods of time [25]. There are a diverse range of stressful conditions in the environment for bacteria, including: temperature, pH, and salt concentration. It has been reported that different *Salmonella* serovars are skilled at adapting to these conditions, due to their ability to survive in temperatures ranging from 2 to 54 °C and pH values from 4.0 to 9.5 [26]. The acid tolerance response of *Salmonella* when passing through the mammalian stomach is presumed to be a factor that enables the pathogen to survive outside the host [27]. Another factor involved in the capability of *Salmonella* to survive environmental stress is the formation of biofilms [28]. Maternity beds are the first contact between the calf and environment; calves can be exposed to different microorganisms at the time of birth, which can compromise their general health. It is essential to keep maternity beds as clean as possible, as preventive measures for young calves, by removing organic materials and sanitizing pen floor, gates, and walls.

Diverse *Salmonella* serovars have been reported in bovines in Mexico, all associated with human infections, the most prevalent being *Salmonella* Montevideo, Anatum, London, Reading, and Typhimurium [22,29,30]. *Salmonella* Montevideo was the most frequent serovar detected in this study, which is consistent with the increasing prevalence of this serovar in America [30]. Given the ubiquity of *Senvironment;terica*, *Salmonella enterica* it is not surprising the high serovar diversity found in the environment, this represents a challenge for the control of *Salmonella* within the farms [20].

The comparison of serogroups using the traditional phenotypic method for serotyping and genome-based prediction, discordance was found in 59% of the samples. Serotyping is the first step to characterizing *Salmonella enterica* and identifying serogroup or serovars for epidemiological purposes, but this does not provide discriminatory subtyping for outbreak investigations. One of the drawbacks of serological methods for the detection of *S. enterica* serovars is that it requires high quality antisera. More than 2600 serovars have been reported, resulting in the combination of 46 O antigens and 114 H antigens [31]. This technique may cause false negative results due to the weak or nonspecific agglutination of O antigen, as observed in this work, underestimating important serogroups that could be associated with dairy farms. Whole genome sequencing represents a powerful tool for *Salmonella enterica* subtyping [32].

The detection of antimicrobial resistance genes was distributed in all isolates analyzed; however, a higher frequency was observed in *Salmonella* Newport, London, and Give, identified in clade III (Figure 3). A high level of intermediate susceptibility against levofloxacin raises questions about the efficiency of this antibiotic in the future. This is possibly due to the point mutation in *parC* gene, which has been reported to confer resistance to ciprofloxacin and nalidixic acid [33]. Mutations in *parC* and *gyrA* genes are highly prevalent among *Salmonella enterica.* The irrational use of antibiotics in agriculture may lead to the accumulation of point mutations in the topoisomerase and DNA gyrase encoding genes and increase the number of fluoroquinolone-resistant pathogens [34].

The phenotypic resistance among the isolates analyzed in this work were higher than those reported by the CDC, which found resistance to ciprofloxacin in 8% of the isolates in 2017 in USA. Countries such as USA, Thailand, and Ethiopia, have also reported a higher frequency of resistance in *Salmonella* isolates compared to the CDC report [35,36,37]. A study conducted in northern California from 2002–2016 on different classes of antibiotics found a reduction in the annual prevalence of antimicrobial resistance for *Salmonella*, and the only antimicrobial with an increasing trend in the annual prevalence was the quinolone drug nalidixic acid [35].

Five plasmid replicons were detected in four isolates. Col(pHAD28) was the most frequent among *Salmonella* serovars. This replicon has been reported in *S. enterica* isolated from pork and chicken and has displayed the presence of the fluoroquinolone resistance *qnrB19* gene [38]. The plasmid IncR detected in *Salmonella* Newport has been associated with clinical multidrug-resistant strains, harboring genes such as *aaadA1*, *sul3*, *tetA(B)*, and *dfrA12* [39]. Studies of plasmid characterization have reported IncQ1 with multiple resistance genes flanked by a repeated region with the *Tn3* Transposase; therefore, the antimicrobial resistance genes could be inserted into the plasmid via transposition. Additionally, the IncQ1 plasmid presents conjugative transfer genes, which represent horizontal mobility potential [40]. Although a low presence of resistant or intermediate resistant isolates were reported in our study, this represents high antimicrobial resistance to tetracycline and ampicillin, compared to reports from countries such as China. Furthermore, in 2020, Wang and colleagues reported that 50% of the isolated *Salmonella* demonstrated multi-drug resistance in adult dairy cows [41]. In another study conducted in poultry farms in Malaysia, a high resistance level was reported against ampicillin and ciprofloxacin (62.5 and 45.8%); this could indicate its misuse in the agricultural sector [42].

A vast number of virulence genes have been described in *Salmonella enterica,* which encoded effectors that induce host infection. These genes can be located in the chromosome, plasmids, pathogenicity islands, or integrated bacteriophages. In this study, several genes associated with fimbrial adherence operons and type II secretion systems 1 and 2 were highly prevalent in the isolates. SPI-1 and 2 encoded membrane-associated type III secretion systems, which are involved in the secretion of effector proteins that modify the functioning of eukaryotic cells and facilitate bacterial pathogenicity inside the cell [43]. The typhoid toxins genes *cdtD*, *pltA*, and *pltB* encode proteins that form a holotoxin, where CdtB and PltA are the active enzymatic units and PltB forms a pore. In six of the analyzed strains these genes were identified, but the *pltB gene* was only present in one isolate, which could indicate the elimination of the virulence activity of this toxin, since PltA/PltB requires the presence for CdtB-mediated toxicity [44]. In this study, 10 SPI were detected, increasing the possibility of horizontal transfer of virulence genes to new clones [45,46]. The identification of more than 170 genes associated with virulence in *S. enterica* isolates represents a public and animal health concern.

The single-nucleotide polymorphism (SNP) phylogeny along with the identification of MLST typing, revealed a high genetic diversity of non-typhoidal *Salmonella* across Mexican states. *Salmonella* Meleagridis ST 463 has been identified in the region of Latin America in pork, beef, and chicken. In Mexico, this serovar is considered one of the most important (in addition to Agona and Anatum); is isolated from the environment, humans, and animals; and has been described as having a multidrug resistance profile [3].

*Salmonella* Montevideo ST 138 has been linked to multistate outbreaks from contaminated food in the U.S., identified back in 2008, associated with vegetables, fruits, and seeds [47]. This sequence type has been identified in healthy cattle isolated from fecal samples and lymph nodes [48]. *Salmonella* Give ST 654 has been isolated from surface water used for aquaculture and crop production in Mexico, along with serovars Meleagridis, Newport, and Anatum [18].

## 4. Materials and Methods

### 4.1. Sample Collection

Samples were collected in a cross-sectional study from calf feces (127), periparturient cow feces (116), and maternity beds (134) from October 2019 to January 2020. Sampling was performed in 55 dairies Mexican farms from Aguascalientes, Baja California, Chihuahua, Coahuila, Durango, Mexico State, Guanajuato, Hidalgo, Jalisco, Queretaro, San Luis Potosi, Tlaxcala, and Zacatecas, where 80% of the total dairy cattle is distributed in Mexico [49] (Figure 4). Calves and periparturient cows were selected randomly and were asymptomatic at the time of sampling. Sample size was determined using a prevalence rate of 30% from previous study in Mexico [17], with a 5% level of significance and based on the sensitivity and specificity of the test at 85%. The minimum number of samples taken from each farm was six, depending on the herd size. Herds were classified by size as follows: less than 500 cows, from 501 to 999, and more than 1000 cows. Samples were taken directly from the rectum of asymptomatic calves from 0 to 60 days of age, asymptomatic close-up cows, and maternity floors with Q-swabs. Samples were stored at 4 °C and sent to the laboratory for microbial isolation.

### 4.2. Salmonella Isolation from Bovine Fecal Samples and Environment

*Salmonella* can be present in small numbers within samples and hidden by large numbers of other *Enterobacteriaceae*. Samples were first enriched in selenite cystine broth (BD, Bacto, Franklin Lakes, NJ, USA) and incubated at 42 °C for 18–24 h [50]. Then, they were grown in XLD agar in the same incubation conditions as enrichment. Black or pink with black-center colonies from each agar were grown in MacConkey agar and lactose-negative colonies were inoculated in Kliger, MIO, and urease medium to identify *Salmonella* via biochemical tests [32]. Strains positive to hydrogen sulfide production with motility, negative ornithine, and urease negative reaction were considered for genus identification via PCR, using a primer set targeting the *ipa*B gene, as described by Fan and colleagues in 2008 [51].

### 4.3. DNA Extraction

The DNA extraction of the samples was performed using the Wizard Genomic DNA purification kit (Promega, Madison, WI, USA), according to manufacturer’s protocol. DNA was quantified in a NanoDrop™ One C (Thermo Fisher Scientific, Waltham, MA, USA)and the extracts were stored at −20 °C until they were used.

### 4.4. Detection of ipaB Gene for Salmonella Genus

Primers used to confirm *Salmonella* genus were according to Fan and colleagues [51]. PCR was carried out in a 15 µL reaction volume containing 1 µL template DNA, 0.2 µL of each primer (0.2 µM), 0.15 µL of dNTPs (100 µM), 0.6 µL of MgCl2 (3 mM), 3 µL of 5X flexi buffer (Promega, Madison, WI, USA), 1 U Taq polymerase (5U/µL) (Promega, Madison, WI, USA), and 9.43 µL of distillated water. The conditions were as follows: initial denaturation at 94 °C for 5 min, followed by 35 cycles of denaturation at 94 °C for 50 s, annealing at 59 °C for 50 s, extension at 72 °C for 100 s, and a final extension at 72 °C for 10 min (modified from Fan et al., 2008) [51]. The PCR product was analyzed using 1.5% agarose gel electrophoresis with pre-added GelGreen dye (Biotium, Fremont, CA, USA) and documented on the transilluminator iBright (Thermo Fisher Scientific, Waltham, MA, USA). A HyperLadder 100 bp (Bioline, London, UK) was used as molecular weight marker. For the quality control of the PCR; *S. enterica* subsp. *enterica* serovar Typhi CDC11; *S. enterica* subsp. *enterica* serovar Enteritidis CDC57; *S. enterica* subsp. *enterica* serovar Typhimurium CDC64; *Shigella flexneri* ATCC29903; and *E. coli* ATCC25922 were used as positive (those from the genus *Salmonella*) and negative (*S. flexneri* and *E. coli*) controls [52]. These were also controls for the biochemical tests.

### 4.5. Antimicrobial Susceptibility Testing

Antimicrobial susceptibility test was conducted via disc diffusion method on Muller–Hinton (MH) (MCD, Mexico State, MX) agar according to the Clinical Laboratory Standards Institute (CLSI) guidelines. *Salmonella* isolates were tested against the following 3 antibiotic groups: penicillin (ampicillin 10 µg), fluoroquinolones (ciprofloxacin 5 µg, levofloxacin 5 µg), and aminoglycosides (amikacin 30 µg, gentamicin 10 µg). These antibiotics were selected due to the critical importance in human and bovines [53]. The diameters of the zone of inhibitions were measured in mm and classified as susceptible, intermediate, or resistant according to CLSI chart for *Enterobacteriaceae* strains [54]. The quality control of the antimicrobial tests was verified with the use of *E. coli* ATTC25922 as a sensitive strain and *P. aeruginosa *ATTC27853 as a resistant strain [55].

### 4.6. Serological Identification of Salmonella enterica O Serogroups

Isolates were serotyped at group level using the Sero-quick group kit (SSI DIAGNOSTICA, Hillerød, Denmark), which identifies seven serogroups (A–G). Strains were grown at 37 °C on a non-selective agar (Muller–Hinton agar, BD Bacto, Franklin Lakes, NJ, USA) and the procedure was performed as described by manufacturer’s protocol. The isolates were serotyped according to the With–Kauffman–Le Minor scheme [1]. For the quality control of the serological identification; *S. enterica* subsp. *enterica* serovar Typhi CDC11; *S. enterica* subsp. *enterica* serovar Enteritidis CDC57; and *S. enterica* subsp. *enterica* serovar Typhimurium CDC64 were used as positive controls and *E. coli* ATCC25922 was used as a negative control [52].

### 4.7. Genome Sequencing and Analysis

Whole-genome sequencing was performed at the Hubbard Center for Genome Studies (University of New Hampshire, Durham, NH, USA) using an Illumina NovaSeq instrument. A paired-end library was constructed using a Nextera DNA library preparation kit (Illumina) to produce 250-bp paired-end reads. The genome assembly metrics and annotation characteristics of the strains are summarized in the Appendix A. The Illumina sequence data were trimmed by Trimmomatic version 0.39 [56]. The quality of the reads was analyzed using the FastQC software version 0.11.8 [57]. Trimmed sequencing reads were assembled using Unicycler version 0.5.0 [58] with default settings. The assembled genomes were annotated via the NCBI Prokaryotic Genome Annotation Pipeline (PGAP).

The analysis of serovars and serogroup was carried out using the EnteroBase page (https://enterobase.warwick.ac.uk/, accessed on 1 October 2023) and the results were compared with SeqSero2 version 1. 1.0 [59] for serotyping prediction. Antimicrobial resistance genes were identified using the genes deposited in the Center for Genomic Epidemiology as templates via the resource of ResFinder 4.1 [60]. Plasmids were identified using PlasmidFinder 2.1 [61] with assembled genomes and a threshold identity of 95%. Virulence genes were identified using the Virulence Factor Database (VFDB) [62]. *Salmonella* pathogenicity islands were detected using the database of the Genomic Epidemiology with SPIFinder 2.0 [63].

## 5. Conclusions

According to the results shown in this study, periparturient cows, calves, and farm environment may represent an important reservoir of different *Salmonella* serogroups. This study indicates that the main source of *Salmonella enterica* are the group maternity areas, where periparturient shedders contaminate and perpetuate the pathogen within the dairy in manure and on concrete surfaces. The isolation of various serovars of *Salmonella* in different samples indicates the wide distribution of this bacterium from animal and human origin and underlines the necessity of surveillance programs in dairy farms. This study raises the necessity of implementing One Health control actions to diminish the prevalence of antimicrobial resistant and virulent pathogens, including *Salmonella*. A longitudinal study with the participation of dairy farms across the country, the analysis of management practices and the use of antimicrobials have the potential to support a much more complete panorama of the bacteria in cattle and in its environment. This study is the first comprehensive characterization of *Salmonella* isolates from dairy farms in multiple states of Mexico, providing an overview of the patterns of infection and genetic characteristics of *Salmonella*.

## Figures and Tables

**Figure 1 antibiotics-12-01662-f001:**
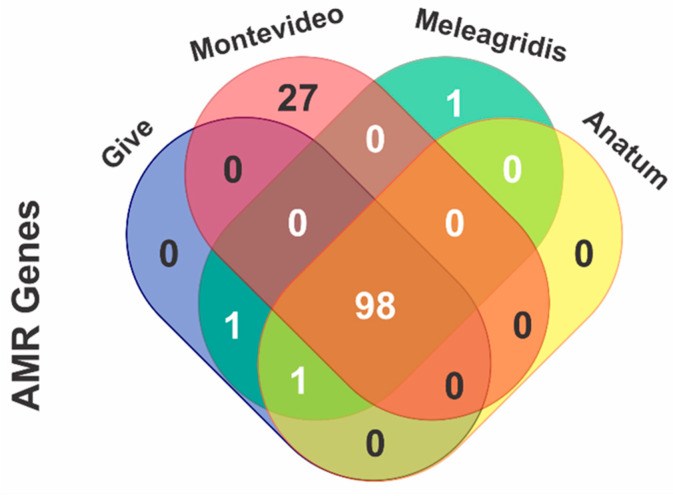
Venn diagram of *Salmonella* isolates for the identification of antimicrobial resistance determinants to the four most prevalent serovars. AMR = antimicrobial resistance.

**Figure 2 antibiotics-12-01662-f002:**
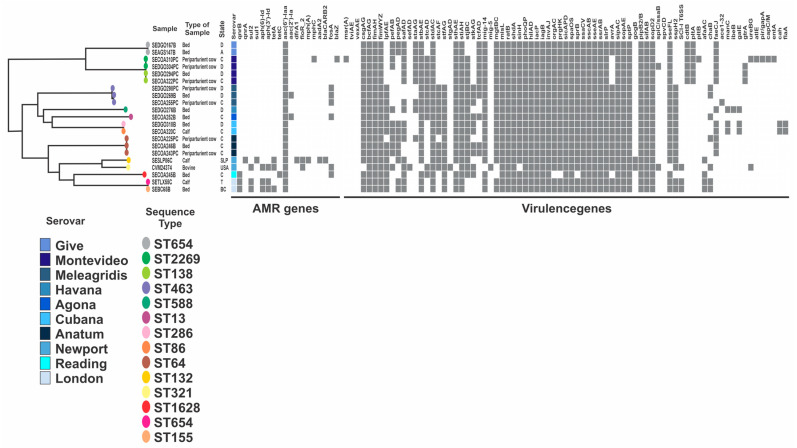
Phylogenetic diversity of *Salmonella* isolates. The maximum-likelihood tree based on core genome SNPs was linked to a binary matrix, which represents the presence and absence of antimicrobial resistance (AMR) and virulence genes. Serovar, sample type, and location for each isolate were also included. Abbreviations; D, Durango; A, Aguascalientes; C, Coahuila; SLP, San Luis Potosí; T, Tlaxcala; BC, Baja California; USA, United States of America.

**Figure 3 antibiotics-12-01662-f003:**
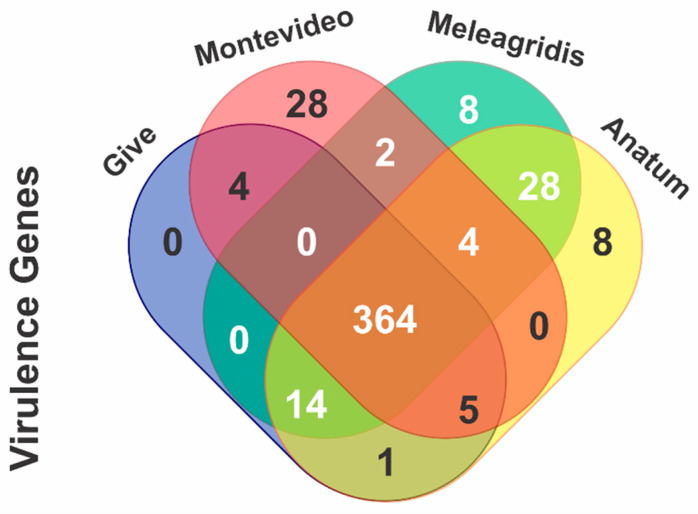
Venn diagram of *Salmonella* isolates for the identification of virulence genes by serovars.

**Figure 4 antibiotics-12-01662-f004:**
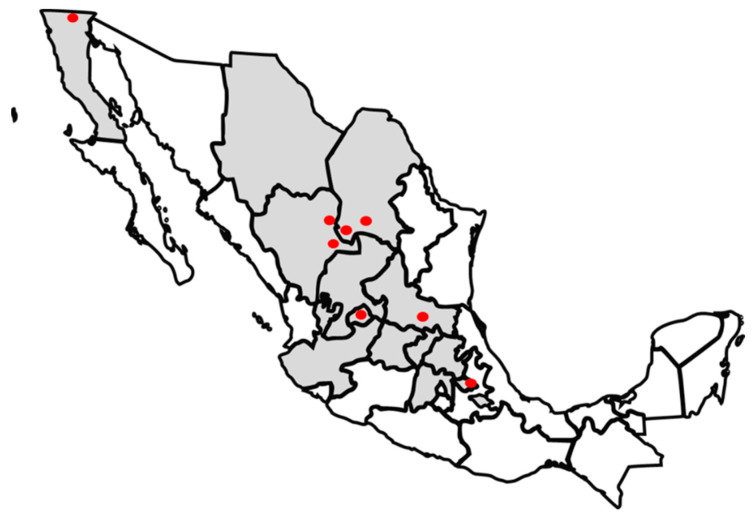
States participating in *Salmonella* study (shown in gray). The farms where the sequenced isolates were obtained are shown in the map in red circles.

**Table 1 antibiotics-12-01662-t001:** Comparison of *Salmonella enterica* serogroup O by serotyping and genome sequencing.

Isolate	Serological Typing	Genome Based in Silico Serotyping
SETLX55C	B	E
SESLP86C	C	C
SEDGO167B	B	E
SECOA243PC	C	E
SECOA252B	C	B
SEDGO294PC	C	C
SEDGO298PC	B	E
SEDGO304PC	C	C
SECOA310PC	C	C
SEDGO318B	D	G
SECOA320C	C	G
SECOA322PC	C	C

**Table 2 antibiotics-12-01662-t002:** Main antibiotic resistance genes distribution among the different *Salmonella* serovars.

Isolate	Serovar	Antibiotic Resistance Genes
SETLX55C	London	*qnr*B19, *sul*2, *aph*(6)-Id, *aph*(3′)-Id, *aac*(6′)-Iaa
SEBC65B	London	*aph*(6)-Id_1, *tet*(A)_6, *aph*(3′)-Id_1, *sul*2_2, *qnr*B19_1, *aac*(6′)-Iaa_1
SESLP86C	Newport	*aac*(6′)-Iaa_1, *dfr*A1_8, *tet*(A)_6, *flo*R_2, *sul*1_15, *mph*(A)_2, *aad*A2_1, *bla*_CARB-2__1, *qnr*A1_1, *qac*E
SEAGS147B	Give	*aac*(6′)-Iaa_1
SEDGO167B	Give	*aac*(6′)-Iaa_1
SECOA225PC	Anatum	*aac*(6′)-Iaa_1
SECOA243PC	Anatum	*aac*(6′)-Iaa_1
SECOA246B	Anatum	*aac*(6′)-Iaa_1
SECOA245B	Reading	*tet*(C)_3, *fos*A7_1, *qnr*B19_1, *aac*(6′)-Iaa_1
SECOA252B	Agona	*aac*(2′)-IIa_1, *fos*A7_1, *aac*(6′)-Iaa_1
SECOA255PC	Meleagridis	*fos*A7_1, *aac*(6′)-Iaa_1
SEDGO269B	Meleagridis	*aac*(6′)-Iaa_1, *fos*A7_1, *aac*(2′)-IIa_1
SEDGO298PC	Meleagridis	*aac*(6′)-Iaa_1, *fos*A7_1
SEDGO276B	Havana	*aac*(6′)-Iaa_1
SEDGO294PC	Montevideo	*aac*(6′)-Iaa_1
SEDGO304PC	Montevideo	*aac*(6′)-Iaa_1
SECOA310PC	Montevideo	*aac*(6′)-Iaa_1, *bla*Z_62, *mph*(C)_2, *msr*(A)_1
SECOA322PC	Montevideo	*aac*(6′)-Iaa_1
SEDGO318B	Cubana	*aac*(6′)-Iaa_1
SECOA320C	Cubana	*aac*(6′)-Iaa_1

**Table 3 antibiotics-12-01662-t003:** Plasmids predicted in *Salmonella* serovars.

Plasmid	Serovar
Col(pHAD28)	London, Reading
IncQ1	London
ColpVC	London
IncR	Newport
Col440I	Reading

## Data Availability

Data are contained within the article and Appendix A. For more information, please contact the authors.

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
