# Peer review of "Serovars, Virulence and Antimicrobial Resistance Genes of Non-Typhoidal Salmonella Strains from Dairy Systems in Mexico"

_antibiotics, 2023, doi:10.3390/antibiotics12121662_

Round 1
Reviewer 1 Report
Comments and Suggestions for Authors
The paper is well designed and needed study. Results are clearly described. I will recommend to address following minor issues for improvement.
1 Although, this particular manuscript is dealing with Salmonella characterization from the dairy animals and their environment, I recommend that authors take into account that major source of salmonella and its transmission to human is poultry meat and other poultry products (add relevant references as mentioned below and give due importance to this aspect in introduction section). Most of the studies throughout the world have emphasize on Salmonellae from the poultry source, hence the characterization of salmonellae from dairy animals and their environment is limited and this piece of work will enhance understanding in this regard.
2. On what criteria, 20 isolates were selected for WGS
3. Use a uniform word through out the article i.e Salmonella spp. or Salmonella genus or Salmonellae
4. Overall, the resistance to antibiotics is still very low which is a good thing. Give a comparison with other geographies where highly resistant Salmonella are reported as well i.e South Asia, China etc, in discussion section.
Some other minor comment are in attached file
Guerrini A, Mescolini G, Roncada P, Tosi G, Raffini E and Frasnelli M, 2021. Seroprevalence and microbiological monitoring in eggs for Salmonella enterica serovar Enteritidis and Salmonella enterica serovar Typhimurium in ornamental chicken flocks in Italy. Pak Vet J, 41(1): 39-44. http://dx.doi.org/10.29261/pakvetj/2020.095
Sadiq S, Ahmad MUD, Chaudhry M, Akbar H, Mushtaq MH, Shehzad F, Hassan S, Khan MUZ, 2021. Molecular epidemiology of zoonotic Salmonella Enteritidis isolated from poultry and human sources by multi locus sequence typing. Pak Vet J, 41(2): 264-268. http://dx.doi.org/10.29261/pakvetj/2020.103

Author Response
- Although, this particular manuscript is dealing with Salmonella characterization from the dairy animals and their environment, I recommend that authors take into account that major source of Salmonella and its transmission to human is poultry meat and other poultry products (add relevant references as mentioned below and give due importance to this aspect in introduction section). Most of the studies throughout the world have emphasize on Salmonellae from the poultry source, hence the characterization of salmonellae from dairy animals and their environment is limited and this piece of work will enhance understanding in this regard.
A: There are many sources of Salmonella, being the most important swine, poultry, and dairy cattle. We add information about the importance of this bacteria in poultry and the impact of Salmonella enterica serovar Enteritidis in animal and human health, comparing this with the main serovars isolated from dairy cattle.
- On what criteria, 20 isolates were selected for WGS
A: The criteria to select the 20 isolates for WGS was added in point 2.3 Serological identification of Salmonella enterica serogroup O, due to from this point on, 20 out of 57 isolations were used for subsequent analyzes.
- Use a uniform word throughout the article i.e Salmonella or Salmonella genus or Salmonellae
A: We change Salmonella spp. to Salmonella to uniform the writing.
- Overall, the resistance to antibiotics is still very low which is a good thing. Give a comparison with other geographies where highly resistant Salmonella are reported as well i.e South Asia, China etc, in discussion section.
A: The discussion about the findings in antibiotic resistance levels were compared geographically with countries that has reported a higher prevalence of resistance such as China and Malaysia as you recommended.
Reviewer 2 Report
Comments and Suggestions for Authors
Thank you for allowing me the opportunity to review the paper titled "Serovars, virulence and antimicrobial resistance genes of non-typhoidal Salmonella strains from dairy systems in Mexico" The paper presents original research and provides valuable insights for the knowledge of the epidemiology and antimicrobial resistance phenomenon of dairy origin Salmonella strains. I support its future processing after appropriate modifications as outlined below:
L17: “Salmonella” – to be consistent, please ensure the italics writing of Salmonella throughout the manuscript
L19: “spread” – not italics
L43: serovars[1] [2] – Please use citations according to the journal requirement
L50: please provide appropriate citations at the end of the sentence/statement
The introduction is well-written and effectively conveys the context and importance of the study. However, the authors must consistently complete this section, and especially lines 60-62, with concrete information and the knowledge gaps of Salmonella infections in livestock, and especially in cattle.
L67: when the authors express overall prevalence values, please insert in brackets the values of the 95% confidence interval
The alignment of the expected findings with the actual results adds credibility to the research.
The discussion section effectively interprets the results and provides a deeper understanding of their significance.
L300: The "Materials and Methods" section is comprehensive, ensuring the study's reproducibility and offering a clear understanding of the techniques used. However, please insert an appropriate reference at the end of the sentence from line 320.
L320: “cystine broth” – please uniformly insert in brackets the manufacturer (production company) name, city, and country for all of the used reagents and devices.
L333: please provide the evidence of using positive and negative controls within PCR reactions
L347: “Antibiotic sensitivity test” – must be “Antimicrobial susceptibility testing”
L350: how were the isolates and the tested antimicrobials selected?
L355: “Salmonella enterica O” – avoid the underlining
L359: no available reference
L379: “our data” – please avoid the personal mode formulations, may sound unprofessional
Within the conclusion section, please highlight the study limitations and indicate further strategies in the approached research area.
Author Response
- L17: “Salmonella” – to be consistent, please ensure the italics writing of Salmonellathroughout the manuscript.
A: We revise the word Salmonella through the entire manuscript to be consistent with the proper writing for bacteria genus.
- L19: “spread” – not italics
A: Thank you for the observation, the change has been made.
- L43: serovars[1] [2] – Please use citations according to the journal requirement
A: Thank you for the observation, the citations have been corrected.
- L50: please provide appropriate citations at the end of the sentence/statement.
A: This sentence corresponds to the Centers for Disease and Control Prevention; it was missing the reference at the end of the sentence. We provide the appropriate citation.
- The introduction is well-written and effectively conveys the context and importance of the study. However, the authors must consistently complete this section, and especially lines 60-62, with concrete information and the knowledge gaps of Salmonella infections in livestock, and especially in cattle.
A: Some gaps in the knowledge of Salmonella presence in dairy cattle are the overall prevalence in animals and the environment, the most common serovars and the sequence types.
- L67: when the authors express overall prevalence values, please insert in brackets the values of the 95% confidence interval.
A: Thank you for the observation, the change has been made.
- The alignment of the expected findings with the actual results adds credibility to the research.
A: Thank you for your comment
- The discussion section effectively interprets the results and provides a deeper understanding of their significance.
A: Thank you for your comment
- L300: The "Materials and Methods" section is comprehensive, ensuring the study's reproducibility and offering a clear understanding of the techniques used. However, please insert an appropriate reference at the end of the sentence from line 320.
A: Thank you for the observation, the change has been made.
- L320: “cystine broth” – please uniformly insert in brackets the manufacturer (production company) name, city, and country for all of the used reagents and devices.
A: We added the city of manufacturing selenite-cystine broth. The name of the company and the country were already described.
- L333: please provide evidence of using positive and negative controls within PCR reactions.
We have added the information about the control strains used for
- L347: “Antibiotic sensitivity test” – must be “Antimicrobial susceptibility testing”
A: Thank you for the observation, the change has been made.
- L350: how were the isolates and the tested antimicrobials selected?
A: Thank you for this observation, the reason why we selected 5 antibiotics belonging to 3 different families was described in the materials and method section. Ampicillin and ciprofloxacin are two of the most important and used antibiotics to treat salmonellosis in animals, we wanted to observe the presence/absence of antimicrobial resistance genes against these antibiotics.
- L355: “Salmonella enterica O” – avoid the underlining.
A: Thank you for the observation, the change has been made.
- L359: no available reference
A: Thank you for your observation, the references has been added.
- L379: “our data” – please avoid the personal mode formulations, may sound unprofessional
A: Thank you for the observation, the change has been made.
- Within the conclusion section, please highlight the study limitations and indicate further strategies in the approached research area.
A: In the conclusion section, we add information about the limitations of the study and further analysis needed to observe a more comprehensive picture of the dynamics of Salmonella in cattle and its environment. The suggestions about the references and the citation according to the journal requirements were addressed.